# Comparative Bonding Analysis of Computer-Aided Design/Computer-Aided Manufacturing Dental Resin Composites with Various Resin Cements

**Yuya Komagata, Yuki Nagamatsu and Hiroshi Ikeda ***

Division of Biomaterials, Department of Oral Functions, Kyushu Dental University, Kitakyushu 803-8580, Japan;
pty_pty_kjm@yahoo.co.jp (Y.K.); yuki-naga@kyu-dent.ac.jp (Y.N.)
*** Correspondence: r16ikeda@fa.kyu-dent.ac.jp

**Abstract:** The use of dental resin composites adapted to computer-aided design/computer-aided manufacturing (CAD/CAM) processes for indirect tooth restoration has increased. A key factor for a successful tooth restoration is the bond between the CAD/CAM composite crown and abutment tooth, achieved using resin-based cement. However, the optimal pairing of the resin cement and CAD/CAM composites remains unclear. This study aimed to identify the optimal combination of a CAD/CAM composite and resin cement for bonding. A commercial methyl methacrylate (MMA)-based resin cement (Super-Bond (SB)) and four other composite-based resin cements (PANAVIA V5; PV, Multilink Automix (MA), ResiCem EX (RC), and RelyX Universal Resin Cement (RX)) were tested experimentally. For the CAD/CAM composites, a commercial polymer-infiltrated ceramic network (PICN)-based composite (VITA ENAMIC (VE)) and two dispersed filler (DF)-based composites (SHOFU BLOCK HC (SH) and CERASMART300 (CE)) were used. Each composite block underwent cutting, polishing, and alumina sandblasting. This was followed by characterization using scanning electron microscopy, inorganic content measurement, surface free energy (SFE) analysis, and shear bond strength (SBS) testing. The results demonstrated that the inorganic content and total SFE of the VE composite were the highest among the examined composites. Furthermore, it bonded highly effectively to all the resin cements. This indicated that PICN-based composites exhibit unique bonding features with resin cements. Additionally, the SBS test results indicated that MMA-based resin cement bonds effectively with both DF- and PICN-based composites. The combination of the PICN-based composite and MMA-based resin cement showed the best bonding performance.

**Keywords:** dental adhesive; composite; methyl methacrylate; bond strength; CAD/CAM; polymer-infiltrated ceramic

## 1. Introduction

The increasing integration of computer-aided design/computer-aided manufacturing (CAD/CAM) systems into dental prosthetic fabrication has revolutionized the field. These systems have substantially streamlined the production of various dental prostheses such as crowns, inlays, onlays, and dentures using CAD/CAM milling machines [1,2]. In addition, CAD/CAM systems can be used to fabricate monolithic restorations and broaden their applicability. This innovation allows for a higher precision in the creation of dental restorations and facilitates faster and more efficient production processes. The advent of CAD/CAM technology in dentistry underscores the significant progress in the industry. It displays potential for future advancements.

The array of materials available through advanced CAD/CAM systems for producing block-shaped restorative dental materials is remarkable [3,4]. Ceramic-based materials include feldspathic porcelain, leucite-reinforced glass, lithium disilicate glass, and zirconia [5,6]. These ceramics enable the fabrication of all-ceramic crown restorations. These restorations are highly sought after because of their exceptional aesthetic quality, robust

mechanical strength, remarkable biocompatibility, and physicochemical properties. These attributes contribute to the long-term functionality of the oral environment. In contrast, resin-based composites are an essential part of advanced dental restorative practice [7]. Resin composites are also known as dental composites or composites. These are tooth-colored materials consisting of an acrylic resin matrix embedded in various types of inorganic fillers. With the growth of CAD/CAM technology, resin composites have been formulated into blocks or discs (referred to as CAD/CAM composites) that are designed specifically for milling restorations within these systems [8]. CAD/CAM composites have grown in popularity owing to their cost-effectiveness, biocompatibility, and superior mechanical and physicochemical properties compared with conventional resin composites used for indirect restorations. These enhancements are largely attributable to a higher degree of matrix resin conversion and increased filler content [9].

Advanced CAD/CAM composites are primarily categorized into two types based on their microstructures: dispersed filler (DF)- and polymer-infiltrated ceramic network (PICN)-based composites [2]. DF-based composites incorporate inorganic fillers into the resin matrix. The DF-based composite for CAD/CAM blocks has a high filler content (60–80%). The resin matrix in the DF-based composite blocks undergoes adequate polymerization under high-temperature and high-pressure conditions during fabrication. Therefore, the mechanical and physicochemical properties of DF-based CAD/CAM composites are superior to those of conventional resin composites. DF-based composites are widely used as restorative materials because of their relatively high flexural strength and machinability [1]. Meanwhile, the microstructure of the PICN-based composites is significantly different from that of the DF-based composites. PICN-based composites feature dual networks composed of ceramic and resin skeletons. As the name indicates, the PICN is a composite material produced by impregnating pre-sintered porous ceramics with resin monomers and polymerizing the resin under high-temperature and high-pressure conditions [10]. Because the PICN composite has a double skeleton comprising a ceramic skeleton and a resin skeleton, the mechanical properties of the PICN are intermediate between those of the resin and ceramics. In particular, PICN-based composites are attracting increasing attention because of their mechanical compatibility with human enamel [11–16]. For example, a commercial PICN-based composite, VITA ENAMIC (VE), consists of a silicate-glass-based ceramic skeleton and methacrylate-based resin skeleton. Its Vickers hardness and elastic modulus are intermediate between those of dentin and enamel. In recent years, PICN-based composites have been used for conventional restorations such as crowns and inlays and for minimal interventions such as occlusal veneers [17], overlays [18], and endocrowns [19].

Tooth preparation design, tooth vitality, and the amount of residual sound tooth structure play vital roles in the long-term success of tooth restoration [18]. Additionally, appropriate bonding of a restorative material to the abutment tooth is crucial. This prevents both fractures and debonding failures [20–22]. CAD/CAM composites that contain large amounts of filler are brittle materials that can crack under relatively small stresses owing to stress concentration. Bonding a composite resin crown to a tooth is important to prevent fractures. This is because crowns integrated with teeth are more resistant to fractures. However, in clinical practice, the debonding failure of crowns fabricated from CAD/CAM composites has been observed [23,24]. These are mainly attributable to the inferior bonding properties of composites with resin cement [25,26]. To address this issue, several surface pretreatment methods have been evaluated to enhance the bond strength between CAD/CAM composites and resin cement. Laser irradiation abrades the surface. This forms a concavo-convex structure that enables effective mechanical interlocking between the CAD/CAM composite and resin cement [27]. In contrast, plasma treatment modifies the surface properties of CAD/CAM composites, thus enhancing the bond strength [28]. Airborne alumina particle abrasion is also known as alumina sandblasting. It abrades the CAD/CAM composite, roughens the surface, and increases the surface area to improve the adhesion [29]. Additionally, the application of an adhesive primer containing a silane coupling agent (known as silane primer) to the surface facilitates covalent or hydrogen

bonding between the CAD/CAM composite and resin cement [30]. Considering factors such as safety, versatility, and usability, alumina sandblasting followed by the application of a silane primer is currently regarded as the most suitable technique for bonding pretreatment of CAD/CAM composite surfaces.

Notwithstanding these advancements, the determination of an appropriate resin cement for optimal bonding with CAD/CAM composites remains an uncharted area of research. In this study, we focused on two types of resin cement: (1) methyl methacrylate (MMA)-based resin cement, which primarily consists of poly(methyl methacrylate) (PMMA) with no filler, and (2) composite-based resin cement, which contains an inorganic filler with a resin matrix. The purpose of this study was to identify the most suitable resin cement for bonding PICN-based and DF-based CAD/CAM composites. The null hypothesis was that the microstructure of the CAD/CAM composite and type of resin cement do not affect the shear bond strength (SBS).

## 2. Materials and Methods

### 2.1. Materials

Table 1 lists the commercially available CAD/CAM composites used in this study, namely, two DF-based composites (HC and CE) and a PICN-based composite (VE). Each CAD/CAM composite block was cut into a 2 mm thick plate using a diamond wheel saw. The surfaces of the plates were polished under dry conditions with emery papers, starting with #400, followed by #600, and finally #1000. The polished plates were then cleaned via ultrasonication in distilled water for 5 min. This was followed by complete drying using an air-blower. Subsequently, the cleaned plates were sandblasted with 50 μm alumina particles using an air-borne particle abrader (Jet Sandblast II, J. Morita, Suita, Japan) under a pressure of 0.2 MPa for 10 s at a distance of 1 cm. The sandblasted plates were subjected to an air-blower to remove the residual alumina particles from the surface. These plates were used for the subsequent experiments.

**Table 1.** CAD/CAM composites. Each composition is according to the manufacturer's information.

| H | Microstructure | Product Name | Manufacturer | Lot Number | Composition |
|---|---|---|---|---|---|
| SH | Dispersed filler | SHOFU BLOCK HC | Shofu, Kyoto, Japan | 0322064 | Silica powder, Zirconium silicate, UDMA, TEGDMA, Micro fumed silica, Pigments |
| CE | Dispersed filler | CERASMART300 | GC, Kyoto, Japan | 2208226 | Balium glass, Silica, Bis-MEPP, UDMA |
| VE | Polymer-infiltrated ceramic network | VITA ENAMIC | Vita Zahnfabrik, Bad Sackingen, Germany | 98162 | $SiO_2$, Al2O$_3$, $Na_2O$, $K_2O$, $B_2O_3$, CaO, $TiO_2$, TEGDMA, UDMA |

UDMA: urethane dimethacrylate; TEGDMA: triethylene glycol dimethacrylate; Bis-MEPP: 2,2-bis(4-methacryloxypolyethoxyphenyl)propane.

A resin cement and adhesive primers were used to bond the CAD/CAM composites. Table 2 details the adhesive systems used in this study. These encompass a methyl methacrylate (MMA)-based resin cement system and four composite-based resin cement systems. These systems are used in conjunction with an adhesive primer and resin cement. The bonding protocols for each resin cement and adhesive primer were followed according to the manufacturer's instructions.

**Table 2.** Resin cements and their corresponding adhesive primers used for bonding with the CAD/CAM composites.

| Adhesive System | Resin Cement (Lot Number) | Adhesive Primer | Manufacturer |
|---|---|---|---|
| SB | Super-Bond (EE22FR) | Super-Bond PZ Primer | Sun Medical, Moriyama, Japan |
| PV | PANAVIA V5 (2E0230) | CLEARFIL CERAMIC PRIMER PLUS | Kuraray Noritake Dental, Tokyo, Japan |
| MA | Multilink Automix (Z02FY6) | Monobond Plus | Ivoclar Vivadent, Schaan, Liechtenstein |
| RC | ResiCem EX (092104) | BeautiBond Xtreme | Shofu, Kyoto, Japan |
| RX | 3M RelyX Universal Resin Cement (9720977) | 3M Scotchbond Universal Plus Adhesive | 3M, Saint Paul, Minnesota, USA |

### 2.2. Scanning Electron Microscopy (SEM)

For the SEM observations, the surfaces of the sandblasted CAD/CAM composites were coated with platinum using a sputtering device. The coated composites were examined via scanning electron microscopy (SEM, JCM-7000, JEOL, Tokyo, Japan) at an accelerating voltage of 10 kV.

### 2.3. Measurement of Filler Contents

The filler content of the CAD/CAM composites was measured using a combustion method as reported earlier [14]. Each sample was weighed using an electric balance (CP225D; Sartorius, Göttingen, Germany). Subsequently, the sample was calcined at 600 °C for 3 h in air to remove all the organic matter. The residual sample ash and inorganic matter (filler) were weighed using an electric balance. The inorganic filler content of the samples was estimated as the ratio of the sample weight before and after calcination.

### 2.4. Surface Free Energy (SFE) Analysis

The SFE of each CAD/CAM composite was determined through the following procedure via contact angle measurements: Initially, the contact angle of each specimen was measured with two test liquids: distilled water and diiodomethane (>99% purity, Kanto Chemical Co., Inc., Tokyo, Japan). Each test liquid's contact angle was obtained five times per sample with each instance involving a droplet volume of 2 μL administered under ambient conditions at a room temperature of $20 \pm 3$ °C, using a dedicated contact angle meter (DMe-211, Kyowa Interface Science Co., Ltd., Saitama, Japan). A new sandblasted sample was used to determine the reliability of each measurement. The contact angle measurements were obtained by capturing an image of the droplet 5 s after it landed on the surface of the specimen. Subsequently, the SFEs of the samples were computed based on the measured contact angles. The calculation involved the utilization of analytical software (FAMAS, Kyowa Interface Science Co., Ltd., Saitama, Japan). The methodology was aligned with the Owens–Wendt theory [31]. The equations governing this theory are as follows:

$$\sqrt{\gamma_{L1}^d \, \gamma_s^d} + \sqrt{\gamma_{L1}^p \, \gamma_s^p} = \frac{\gamma_{L1}^{total}(1 + cos\theta_{L1})}{2} \tag{1}$$

$$\sqrt{\gamma_{L2}^d \, \gamma_s^d} + \sqrt{\gamma_{L2}^p \, \gamma_s^p} = \frac{\gamma_{L2}^{total}(1 + cos\theta_{L2})}{2} \tag{2}$$

$$\gamma^{total} = \gamma^d + \gamma^p \tag{3}$$

where the subscripts L1 and L2 indicate the test liquids (water and diiodomethane, respectively); $\gamma^{total}$, $\gamma^p$, and $\gamma^d$ are the total SFE, polar (hydrogen) component of SFE, and dispersive component of SFE, respectively, for the examined composites (or control samples); and $\gamma_L^{total}$, $\gamma_L^p$, and $\gamma_L^d$ are the total SFE, polar component of SFE, and dispersed component of SFE, respectively, for the test liquids (water and diiodomethane). The SFE

values of the test liquids used were the following previously reported ones [31]: $\gamma_{L1}^{total} = 72.8 \, mN/m$, $\gamma_{L1}^{p} = 51.0 \, mN/m$, and $\gamma_{L1}^{d} = 21.8 \, mN/m$ for water, and $\gamma_{L2}^{total} = 50.8 \, mN/m$, $\gamma_{L2}^{p} = 1.3 \, mN/m$, and $\gamma_{L2}^{d} = 49.5 \, mN/m$ for diiodomethane. $\theta$ is the measured contact angle of the test liquids.

### 2.5. Shear Bond Strength (SBS) Test

The bond strength of each CAD/CAM composite with each resin cement was determined using the SBS test reported in our previous study [26]. The sandblasted CAD/CAM composite plates were secured to a Teflon tube (height = 5 mm, inner diameter = 5 mm) using double-coated tape to ensure a constant bonding area of 19.6 mm². A primer was applied to the bonding area of the CAD/CAM composite surface. Subsequently, the resin cement was loaded onto the composite surface and cured according to the manufacturer's instructions. The cement-bonded samples were then immersed in distilled water at 37 °C for 1 d. These samples were designated as the "initial group". Additionally, samples were subjected to 20,000 thermal cycles by alternately immersing these in 5 °C and 55 °C water baths for 60 s each. These samples were designated as the "thermocycling group".

The SBS tests were performed on both initial and thermocycling groups using a universal testing machine (AGS-H; Shimadzu Corp., Kyoto, Japan). A shear load was applied to the interface between the CAD/CAM composite and resin cement using a knife-edge-shaped apparatus until failure occurred. The SBS value was calculated by dividing the measured maximum applied force by the adhesion area.

After the SBS test, the fractured interface between the CAD/CAM composite and resin cement was inspected to determine its failure mode. The failure mode was categorized into three types: adhesive failure at the cement–composite interface, cohesive failure within the composite, and mixed failure consisting of both cohesive and adhesive types.

### 2.6. Statistical Analysis

Statistical analyses of SBS, SFE, and filler content were performed using EZR software (Jichi Medical University, Saitama, Japan). The Kolmogorov–Smirnov test was used to assess the normality of data distribution. Given that the results indicated a non-normal data distribution, we used the non-parametric Steel–Dwass test for multiple comparisons. In all the analyses, the threshold for statistical significance (*p*-value) was set at 0.05.

## 3. Results

The SEM images of the sandblasted composites presented in Figure 1 demonstrate that each composite surface was roughened by the alumina sandblasting, thereby forming random microgrooves. Figure 2 illustrates the inorganic content of the composites as determined by the calcination method. The content follows the order SH ($63.87 \pm 0.16\%$) < CE ($74.95 \pm 0.40\%$) < VE ($86.46 \pm 0.22\%$). This inorganic content was aligned with the manufacturer's information for each product.

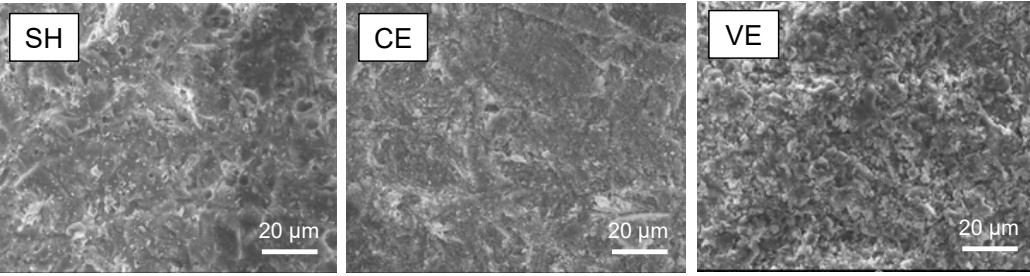

**Figure 1.** SEM images of the sandblasted CAD/CAM composites.

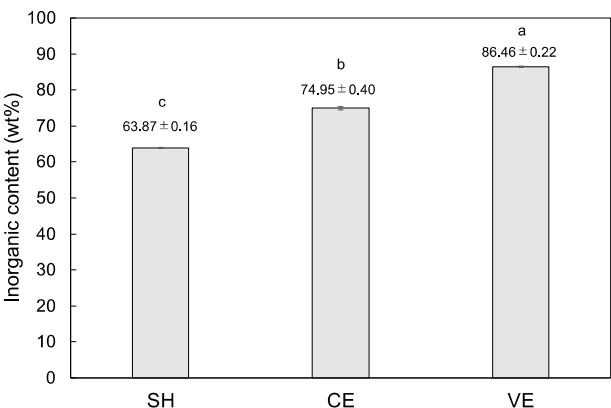

**Figure 2.** Inorganic content of the CAD/CAM composites. The different alphabet letters in the figure represent statistically significant differences between the groups (*p* < 0.05, Steel–Dwass test).

The SFE analysis results of the composites are shown in Figure 3. The polar components of the VE composite are significantly higher than those of the SH and CE composites. The dispersive component of the VE composite was lower than those of the SH and CE composites. The VE composite exhibited the highest total SFE (which consists of the polar and dispersive components) among the examined composites.

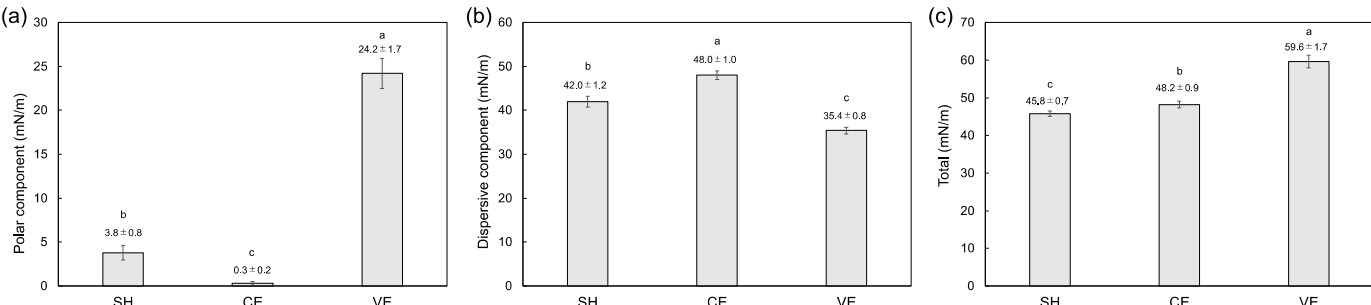

**Figure 3.** SFE of the sandblasted CAD/CAM composites, (**a**) polar component of SFE, (**b**) dispersive component of SFE, and (**c**) total SFE. The different alphabet letters in the figure represent statistically significant differences between the groups (*p* < 0.05, Tukey's test).

Table 3 outlines the SBS between each resin cement and composite. It includes the results of the statistical comparisons of the SBSs. The statistical analysis revealed that the type of cement affected the SBS. Upon comparing the SBS across the SH composite, the Super-Bond (SB) cement displayed the highest value in both initial and thermocycling groups. Similarly, for the CE composite, the SB cement consistently exhibited the highest values. For the VE composite, the PANAVIA V5 (PV) cement had the highest value among the initial groups, whereas the SB cement had the highest SBS among the thermocycling groups. These results indicate that the SB cement displayed the best performance in bonding with each composite. The statistical analysis also revealed that the type of composite affected the SBS. For the cement types, the SBS followed the order SH < CE < VE. This indicates that the VE composite performed the best for bonding with cement.

**Table 3.** Mean and standard deviation of SBS between each adhesive system and each CAD/CAM composite. The uppercase letters indicate significant differences between groups in rows, whereas the lowercase letters represent significant differences between groups in columns ($p < 0.05$, Steel–Dwass test, $n = 11$).

| Adhesive | SH | | CE | | VE | |
|---|---|---|---|---|---|---|
| | Initial | Thermocycling | Initial | Thermocycling | Initial | Thermocycling |
| SB | 17.9 ± 3.2 A, ab | 19.2 ± 2.0 A, ab | 20.9 ± 4.2 A, a | 18.9 ± 3.9 A, ab | 21.6 ± 5.2 B, a | 15.0 ± 3.4 A, b |
| PV | 4.4 ± 0.8 D, cd | 3.4 ± 1.1 B, d | 10.5 ± 1.3 B, b | 6.3 ± 0.5 B, bcd | 30.9 ± 9.9 A, a | 9.5 ± 3.5 AB, bc |
| MA | 6.6 ± 0.7 CD, cd | 2.7 ± 1.9 B, d | 10.3 ± 1.3 BC, bc | 8.0 ± 1.4 B, bc | 23.6 ± 6.7 AB, a | 12.5 ± 7.0 AB, b |
| RC | 9.9 ± 2.1 B, b | 2.5 ± 1.5 B, c | 7.7 ± 1.3 C, b | 7.4 ± 1.1 B, b | 28.1 ± 7.7 AB, a | 7.8 ± 3.8 B, b |
| RX | 8.0 ± 1.8 BC, c | 1.9 ± 0.7 B, d | 9.9 ± 1.7 BC, bc | 8.5 ± 2.6 B, c | 31.7 ± 3.6 A, a | 14.3 ± 8.0 AB, b |

The SBS results were supported by the failure mode analysis. Table 4 lists the failure modes of the samples after the SBS tests. It indicates that cohesive failure more strongly bonded the resin cement with the composite than adhesive failure. Cohesive failure was observed only in the SB cement for all the composites in both initial and thermocycling groups. This indicates that the SB cement bonded well to each composite even after thermocycling. Focusing on the composite types, cohesive failure was most frequently observed in the VE composite. This implied that it bonded effectively with all the cements.

**Table 4.** Failure modes (Adhesive/Mix/Cohesive) of the SBS-tested samples.

| Adhesive | SH | | CE | | VE | |
|---|---|---|---|---|---|---|
| | Initial | Thermocycling | Initial | Thermocycling | Initial | Thermocycling |
| SB | 0/0/11 | 0/0/11 | 0/0/11 | 0/0/11 | 0/0/11 | 0/0/11 |
| PV | 11/0/0 | 9/2/0 | 11/0/0 | 11/0/0 | 0/0/11 | 2/5/4 |
| MA | 11/0/0 | 8/3/0 | 6/4/1 | 3/8/0 | 0/0/11 | 0/8/3 |
| RC | 3/7/1 | 10/1/0 | 10/1/0 | 10/1/0 | 0/0/11 | 1/9/1 |
| RX | 3/8/0 | 11/0/0 | 6/5/0 | 4/7/0 | 0/0/11 | 1/6/4 |

## 4. Discussion

The relationship between the microstructure of the CAD/CAM composites and their bonding characteristics remains ambiguous in contemporary dental prosthetic research. The ambiguity on this issue was the primary motivation for the current investigation. It sought to determine the bond strength between CAD/CAM composites and resin cements, with a specific focus on various types of commercially available composites and resin cements. In this study, we selected three commercial CAD/CAM composite blocks to represent the range of materials commonly used in the field. These included two DF-based composites, labelled SH and CE, and a PICN-based composite, denoted as VE. These composite options enabled us to analyze the interactions of both the major types of CAD/CAM composites with different types of resin cements. With respect to the resin cements, we used a commercial MMA-based resin cement, identified as SB (Super-Bond), and four composite-based resin cements, namely, PV (PANAVIA V5), MA (Multilink Automix), RC (ResiCem EX), and RX (3M RelyX Universal Resin Cement). An appropriate silane-containing primer was incorporated into the bonding process to facilitate effective adhesion between the composites and cement. To test the bond strength, SBS tests were performed for each pairing of composite and cement. The tests revealed observable differences in the SBS values across various combinations of materials. This disparity in results demonstrates the influence of both the type of resin cement and the microstructure of the CAD/CAM composite on

the SBS. Therefore, we reject our null hypothesis. This highlights the significance of these variables in influencing the bond strength of dental restorations.

Alumina sandblasting is conventionally used as a surface pretreatment approach to enhance the bonding of various restorative materials including metal-based materials (e.g., gold alloys and titanium), resin-based materials (e.g., PMMA and resin composites), and polycrystalline ceramics (e.g., zirconia) [32]. Alumina sandblasting is considered a practical method in dental clinics because of its safety, convenience, and ease of handling. In addition, it is an effective surface pretreatment method for increasing the bond strength of resin cements [33,34] because it yields increased surface roughness and SFE. These, in turn, contribute to improved bonding with adhesives. This study demonstrated that alumina sandblasting successfully roughened the composite surface. Moreover, the increased surface roughness resulted in enhanced mechanical interlocking at the composite–adhesive interface.

The results of the SBS tests highlighted the significant influence of the composite microstructure on the bond strength between the composite and resin cement. The SBS of the PICN-based composite (VE) exceeded that of the DF-based composites (SH and CE). These observations were corroborated by the failure mode analysis. The analysis revealed that the PICN-based composite predominantly demonstrated cohesive failure modes, which were significantly more prevalent than those observed for the DF-based composites. The implication of these SBS test results is the potential superiority of the PICN-based composite in terms of bonding properties compared with the DF-based composites. Our present observations align with the results of prior research [26,35]. This demonstrates that resin cements with a silane primer perform more effectively with PICN-based composites than with DF-based composites. This assertion validates the SFE results. The polar component of the SFE of the PICN-based composite was significantly higher than that of the DF-based composite. According to the Owens–Wendt theory, the polar component of SFE originates from polar functional groups capable of forming hydrogen bonds such as Si-OH groups. Consequently, we determined that the surface of the PICN-based composite displayed a relatively larger number of Si-OH groups than that of the DF-based composites. These Si-OH groups on the composite surface can react with the silane coupling agent present in the silane primer. This would enhance the bond strength between the composite surface and resin cement. The sandblasting process was applied to increase the surface area of both the PICN-based and DF-based composites. This resulted in a corresponding increase in the SFE of the composites. It is also worth mentioning that the filler content within the composites could potentially affect the bond strength of the resin cement via a silane coupling agent. Because the effect of the silane coupling agent increased with an increase in the filler content, the filler content in our test composites followed the order SH < CE < VE. This order is consistent with the SBS results and further reinforces our observations.

The SBS results revealed that the MMA-based resin cement presented a superior bonding performance with the CAD/CAM composites, compared with the composite-based resin cements. This aligns with previous studies [36]. Therein, the MMA-based resin cement Super-Bond adhesive system demonstrated a higher bond strength than the composite-based resin cement ResiCem adhesive system. A feasible explanation for this phenomenon involves the formation of a semi-interpenetrating polymer network (semi-IPN) structure at the interface between the MMA-based resin cement and the CAD/CAM composite [37–39]. The structure comprises a macromolecular-level polymer blend in which the polymer chains of a linear polymer infiltrate another polymer matrix. It is known to enhance the bond strength between different polymer resins via mechanical interlocking [37]. In this study, it was postulated that the relatively small MMA molecules infiltrate the resin matrix of the CAD/CAM composite and subsequently polymerize to PMMA. This results in the formation of the semi-IPN structure at the interface. This structure potentially enhances the bond strength between the MMA-based resin cement and the CAD/CAM composite. Furthermore, the use of an MMA-containing silane primer in conjunction with an MMA-based resin cement is considered to facilitate the semi-IPN structure formation.

This is because the MMA present in the primer can penetrate the resin matrix. This supposition is supported by reports stating that an MMA-containing primer enhances the bonding between CAD/CAM composites and resin cement [27,30,40,41] and that the MMA monomer can infiltrate the resin matrix in resin composites [30,38,41,42]. In contrast, the composite-based resin cements employed in this study contained relatively large molecules such as triethylene glycol dimethacrylate (TEGDMA), urethane dimethacrylate (UDMA), and bisphenol A glycidyl methacrylate (Bis-GMA). These do not have the capability to infiltrate another polymer network or form a semi-IPN structure. Therefore, the SBS decreased significantly after thermocycling because of the deficiency of mechanical interlocking by the semi-IPN structure. Another consideration is the wettability of the resin cement. As shown in the SEM images, the surfaces of the CAD/CAM composites featured grooves of various sizes formed by sandblasting. Owing to its relatively low viscosity and high wettability, MMA-based resin cement can infiltrate both wide and narrow grooves. This infiltration results in mechanical interlocking at the interface when the MMA-based resin cement is cured. In contrast, composite-based resin cements, which contain numerous ceramic fillers (particles), are incapable of infiltrating narrow grooves on CAD/CAM composite surfaces.

The current study was limited to the use of a single PICN-based composite owing to the restricted commercial availability of this type of composite. Currently, the only available brand is ENAMIC. Future studies should aim to test the bond strength using a diverse range of PICN-based composites including noncommercial prototypes [15,16,43,44]. The MMA resin cement used in this study was restricted to one brand. Therefore, future studies should consider the use of other brands or prototypes of MMA-based resin cements. Moreover, although the present experiments provided valuable in vitro insights, it is crucial to acknowledge that these laboratory conditions may not fully replicate the complexities of the oral environment. Consequently, the next step in evaluating the performance of these materials involves clinical studies to verify their efficacy and durability in oral settings. Through such investigations, the bond strength characteristics of these materials and their potential applications in dental practice can be understood more comprehensively.

## 5. Conclusions

This study was focused on evaluating the bonding performance between CAD/CAM composites (namely, PICN- and DF-based composites) and resin cements (specifically MMA- and composite-based resin cements). With due consideration of the limitations of this study, we drew the following conclusions:

1.  When resin cement was used with a silane coupling agent, the PICN-based CAD/CAM composite displayed a superior bond strength compared with its DF-based CAD/CAM counterpart. These observations underscore the potential advantages of PICN-based CAD/CAM composites for dental restorations.
2.  Irrespective of whether a PICN- or DF-based CAD/CAM composite was used, a higher bond strength was observed with the MMA-based resin cement than with the composite-based resin cements. This indicates that the MMA-based resin cement may provide superior bonding properties and, thereby, facilitate better dental restorative procedures.

These conclusions highlight the advantages of PICN-based composites and MMA-based resin cements for achieving superior bond strength. This is crucial for the long-term durability and efficacy of dental restorations.

**Author Contributions:** Conceptualization, H.I. and Y.K.; data curation, Y.K. and H.I.; formal analysis, Y.K. and Y.N.; funding acquisition, H.I.; investigation, Y.K. and Y.N.; methodology, Y.K.; project administration, H.I.; resources, H.I.; software, Y.K. and Y.N.; supervision, H.I.; validation, Y.N. and H.I.; visualization, Y.K. and H.I.; writing—original draft, Y.K. and Y.N.; writing—review and editing, H.I. All authors have read and agreed to the published version of the manuscript.

**Funding:** This research was funded by JSPS KAKENHI (grant number 23K09236).



**Data Availability Statement:** The data presented in this study are available upon request from the corresponding author.

**Conflicts of Interest:** The authors declare no conflict of interest.

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
