# Peer review of "Comparative Bonding Analysis of Computer-Aided Design/Computer-Aided Manufacturing Dental Resin Composites with Various Resin Cements"

_jcs, doi:10.3390/jcs7100418_

Round 1

Reviewer 1 Report

Dear Authors

I appreciated the quality of the paper. Methods, statistical analysis and results are well detailed and presented.

I suggest to add more details regarding the methods of application of the various resin cements. (application of primer, air blowing, bonding, photoactivation, automixing,...). Readers should be aware of the different materials used in order to understand differences.

In figures 1,2,3 and Table 1, the abbreviations used for the composites are not the same.

Best regards

Author Response

We would like to express our appreciation to all the reviewers for their insightful comments regarding our paper. The comments and suggestions have helped us to improve the paper. The reviewers’ comments and our responses are presented below.

Comments from reviewer #1

I appreciated the quality of the paper. Methods, statistical analysis and results are well detailed and presented. I suggest to add more details regarding the methods of application of the various resin cements. (application of primer, air blowing, bonding, photoactivation, automixing,...). Readers should be aware of the different materials used in order to understand differences. In figures 1,2,3 and Table 1, the abbreviations used for the composites are not the same.

Response to Reviewer #1

The bonding procedure depends on the resin cement brand. The description will be large and complex. In order to avoid unreadable text, the following sentences were written in the text.” The bonding protocols for each resin cement and adhesive primer were followed according to the manufacturer’s instructions.”

The abbreviations have been corrected in the text.

Reviewer 2 Report

The paper “Comparative Bonding Analysis of CAD/CAM Dental Resin Composites with Various Resin Cements” aims to identify the most suitable resin cement for bonding PICN-based and DF-based CAD/CAM composites. 

The article covers a very interesting topic. Major improvements shall be provided before possible publication.

The authors wrote:

“As an acknowledgment of their 52 effectiveness and credibility, CAD/CAM composites are included in health insurance coverage for both premolar and molar restorations in Japan [9] “

The reviewer does not see why this sentence related to insurances should be included in the manuscript. Cad/Cam composites are reliable materials despite their use in insurance plans.

The authors wrote:

“Previous studies have reported that a PICN-based composite composed of a nanosized silica skeleton exhibited mechanical compatibility with human teeth in terms of hardness and flexural modulus [14,15]. Eldafrawy et al. developed a functionally graded PICN block with a gradient of mechanical and optical properties through the thickness of the block. It was designed to replicate 83 tooth properties [16] ”

The authors could remove these sentences from the introduction or move them (and discuss) in the Discussion

Line 122:

The authors wrote:

“The surfaces of the plates were polished with emery papers up to #1000.”

Please list the other grains, time of application and if water was or was not used.

Lines 124-6:

The authors wrote:

“Subsequently, the cleaned plates were sandblasted with 50 μm alumina particles using an air-borne particle abrader (Jet Sandblast II, J. Morita, Suita, Japan) under a pressure of 0.2 MPa for 10 s.”

Please report sand-blasting distance in cm.

Table 1:

please report lot number of the investigated materials

Table 2:

please report lot number of the investigated materials

Lines 197-200:

The authors wrote:

“A shear load was applied to the interface between the CAD/CAM composite and resin cement using a knife-edge- shaped apparatus until failure occurred. The SBS value was calculated by dividing the measured maximum applied force by the adhesion area.”

Please report the load, time and all the data in order to make this experiment reproduced by others.

Lines 87-92

The authors could also outline that the bond strength (analysed in this paper) depends (in long-term!) also on clinical factors such as preparation design, tooth structure, vitality and structural analysis. The authors could add the following sentence with the following reference:

“Nevertheless, successful long-term restorations depends on several factors, such as preparation design, vitality of the tooth, residual sound tooth structure, etc.”

Reference: Comba A, Baldi A, Carossa M, et al. Post-Fatigue Fracture Resistance of Lithium Disilicate and Polymer-Infiltrated Ceramic Network Indirect Restorations over Endodontically-Treated Molars with Different Preparation Designs: An In-Vitro Study. Polymers (Basel). 2022;14(23):5084. Published 2022 Nov 23. doi:10.3390/polym14235084

https://doi.org/10.3390/polym14235084

Author Response

Comments from reviewer #2

The paper “Comparative Bonding Analysis of CAD/CAM Dental Resin Composites with Various Resin Cements” aims to identify the most suitable resin cement for bonding PICN-based and DF-based CAD/CAM composites.

The article covers a very interesting topic. Major improvements shall be provided before possible publication.

Q.2-1: The authors wrote:

“As an acknowledgment of their 52 effectiveness and credibility, CAD/CAM composites are included in health insurance coverage for both premolar and molar restorations in Japan [9] “

The reviewer does not see why this sentence related to insurances should be included in the manuscript. Cad/Cam composites are reliable materials despite their use in insurance plans.

A.2-1: This sentence has been eliminated from the revised manuscript.

Q.2-2: The authors wrote:

“Previous studies have reported that a PICN-based composite composed of a nanosized silica skeleton exhibited mechanical compatibility with human teeth in terms of hardness and flexural modulus [14,15]. Eldafrawy et al. developed a functionally graded PICN block with a gradient of mechanical and optical properties through the thickness of the block. It was designed to replicate 83 tooth properties [16] ”

The authors could remove these sentences from the introduction or move them (and discuss) in the Discussion.

A.2-2: This sentence has been eliminated from the introduction in the revised manuscript.

Q.2-3: Line 122:

The authors wrote:

“The surfaces of the plates were polished with emery papers up to #1000.”

Please list the other grains, time of application and if water was or was not used.

A.2-3: The sentence has been modified and inserted into the revised manuscript.

“The surfaces of the plates were polished under dry conditions with emery papers, starting with #400, followed by #600, and finally #1000.”

Q.2-4: Lines 124-6:

The authors wrote:

“Subsequently, the cleaned plates were sandblasted with 50 μm alumina particles using an air-borne particle abrader (Jet Sandblast II, J. Morita, Suita, Japan) under a pressure of 0.2 MPa for 10 s.”

Please report sand-blasting distance in cm.

A.2-4: The sentence has been modified as follow.

“the cleaned plates were sandblasted with 50 μm alumina particles using an air-borne particle abrader (Jet Sandblast II, J. Morita, Suita, Japan) under a pressure of 0.2 MPa for 10 s at a distance of 1 cm.”

Q.2-5: Table 1:

please report lot number of the investigated materials

A.2-5: The lot numbers have been added in the table.

Q.2-6: Table 2:

please report lot number of the investigated materials

A.2-6: The lot numbers have been added in the table.

Q.2-7: Lines 197-200:

The authors wrote:

“A shear load was applied to the interface between the CAD/CAM composite and resin cement using a knife-edge-shaped apparatus until failure occurred. The SBS value was calculated by dividing the measured maximum applied force by the adhesion area.”

Please report the load, time and all the data in order to make this experiment reproduced by others.

A.2-7: As you suggested, row data is important for reproducibility of the experiment. However, if all the data such as stress-strain curves are included in the manuscript, the amount will be enormous. Therefore, only the important value (maximum stress) is listed in this paper. In general, almost published papers study on bonding properties of dental materials are written in the same format.

Q.2-8: Lines 87-92

The authors could also outline that the bond strength (analysed in this paper) depends (in long-term!) also on clinical factors such as preparation design, tooth structure, vitality and structural analysis. The authors could add the following sentence with the following reference:

“Nevertheless, successful long-term restorations depends on several factors, such as preparation design, vitality of the tooth, residual sound tooth structure, etc.”

Reference: Comba A, Baldi A, Carossa M, et al. Post-Fatigue Fracture Resistance of Lithium Disilicate and Polymer-Infiltrated Ceramic Network Indirect Restorations over Endodontically-Treated Molars with Different Preparation Designs: An In-Vitro Study. Polymers (Basel). 2022;14(23):5084. Published 2022 Nov 23. doi:10.3390/polym14235084

A.2-8: This sentence has been modified and inserted in the text with the above reference.

Round 2

Reviewer 1 Report

Dear Authors,

I appreciate the answers you provided to the remarks. 

Best regards 

Reviewer 2 Report

All comments have been addressed.